# Swelling, Rupture and Endosomal Escape of Biological Nanoparticles Per Se and Those Fused with Liposomes in Acidic Environment

**DOI:** 10.3390/pharmaceutics16050667

**Published:** 2024-05-16

**Authors:** Natalia Ponomareva, Sergey Brezgin, Ivan Karandashov, Anastasiya Kostyusheva, Polina Demina, Olga Slatinskaya, Ekaterina Bayurova, Denis Silachev, Vadim S. Pokrovsky, Vladimir Gegechkori, Evgeny Khaydukov, Georgy Maksimov, Anastasia Frolova, Ilya Gordeychuk, Andrey A. Zamyatnin Jr., Vladimir Chulanov, Alessandro Parodi, Dmitry Kostyushev

**Affiliations:** 1Laboratory of Genetic Technologies, Martsinovsky Institute of Medical Parasitology, Tropical and Vector-Borne Diseases, First Moscow State Medical University (Sechenov University), 119991 Moscow, Russia; ivan.karandashov@gmail.com (I.K.); kostyusheva_ap@mail.ru (A.K.); vladimir@chulanov.ru (V.C.); dkostushev@gmail.com (D.K.); 2Division of Biotechnology, Sirius University of Science and Technology, 354340 Sochi, Russia; vadimpokrovsky@yandex.ru (V.S.P.); frolanasta@gmail.com (A.F.); zamyat@belozersky.msu.ru (A.A.Z.J.); aparodi.sechenovuniversity@gmail.com (A.P.); 3Department of Pharmaceutical and Toxicological Chemistry, Sechenov First Moscow State Medical University, 119146 Moscow, Russia; gegechkori_v_i@staff.sechenov.ru; 4Institute of Physics, Technology, and Informational Systems, Moscow Pedagogical State University, Malaya Pirogovskaya St. 1, 119435 Moscow, Russia; polidemina1207@yandex.ru (P.D.); khaydukov@mail.ru (E.K.); 5National Research Centre “Kurchatov Institute”, Akademika Kurchatova Sq. 1, 123182 Moscow, Russia; 6Faculty of Biology, Lomonosov Moscow State University, 119991 Moscow, Russia; slatolya@mail.ru (O.S.); gmaksimov@mail.ru (G.M.); 7Chumakov Federal Scientific Center for Research and Development of Immunobiological Products, Russian Academy of Sciences (Polio Institute), 108819 Moscow, Russia; bayurova_eo@chumakovs.su (E.B.); gordeychuk_iv@chumakovs.su (I.G.); 8V.I. Kulakov National Medical Research Center of Obstetrics, Gynecology and Perinatology, 117997 Moscow, Russia; silachevdn@belozersky.msu.ru; 9A.N. Belozersky Institute of Physico-Chemical Biology, Lomonosov Moscow State University, 119991 Moscow, Russia; 10Blokhin National Medical Research Center of Oncology, 115478 Moscow, Russia; 11Department of Biochemistry, People’s Friendship University, 117198 Moscow, Russia; 12Institute of Molecular Medicine, Sechenov First Moscow State Medical University (Sechenov University), 119435 Moscow, Russia; 13Faculty of Bioengineering and Bioinformatics, Lomonosov Moscow State University, 119234 Moscow, Russia; 14Department of Infectious Diseases, First Moscow State Medical University (Sechenov University), 119991 Moscow, Russia

**Keywords:** exosomes, proton sponge, RNA, CRISPR/Cas, small molecules, cancer, gene editing, vaccines, biological barriers, surface-enhanced Raman spectroscopy, nanoparticle-tracking assay, dynamic light scattering

## Abstract

Biological nanoparticles (NPs), such as extracellular vesicles (EVs), exosome-mimetic nanovesicles (EMNVs) and nanoghosts (NGs), are perspective non-viral delivery vehicles for all types of therapeutic cargo. Biological NPs are renowned for their exceptional biocompatibility and safety, alongside their ease of functionalization, but a significant challenge arises when attempting to load therapeutic payloads, such as nucleic acids (NAs). One effective strategy involves fusing biological NPs with liposomes loaded with NAs, resulting in hybrid carriers that offer the benefits of both biological NPs and the capacity for high cargo loads. Despite their unique parameters, one of the major issues of virtually any nanoformulation is the ability to escape degradation in the compartment of endosomes and lysosomes which determines the overall efficiency of nanotherapeutics. In this study, we fabricated all major types of biological and hybrid NPs and studied their response to the acidic environment observed in the endolysosomal compartment. In this study, we show that EMNVs display increased protonation and swelling relative to EVs and NGs in an acidic environment. Furthermore, the hybrid NPs exhibit an even greater response compared to EMNVs. Short-term incubation of EMNVs in acidic pH corresponding to late endosomes and lysosomes again induces protonation and swelling, whereas hybrid NPs are ruptured, resulting in the decline in their quantities. Our findings demonstrate that in an acidic environment, there is enhanced rupture and release of vesicular cargo observed in hybrid EMNVs that are fused with liposomes compared to EMNVs alone. This was confirmed through PAGE electrophoresis analysis of mCherry protein loaded into nanoparticles. In vitro analysis of NPs colocalization with lysosomes in HepG2 cells demonstrated that EMNVs mostly avoid the endolysosomal compartment, whereas hybrid NPs escape it over time. To conclude, (1) hybrid biological NPs fused with liposomes appear more efficient in the endolysosomal escape via the mechanism of proton sponge-associated scavenging of protons by NPs, influx of counterions and water, and rupture of endo/lysosomes, but (2) EMNVs are much more efficient than hybrid NPs in actually avoiding the endolysosomal compartment in human cells. These results reveal biochemical differences across four major types of biological and hybrid NPs and indicate that EMNVs are more efficient in escaping or avoiding the endolysosomal compartment.

## 1. Introduction

In recent decades, advances in nanotechnology have enabled the targeted delivery of many therapeutics, improving their efficiency, safety and pharmacokinetics properties. Several formulations, including lipid-based (Doxil, Caelyx, DaunoXome, Myocet, etc.), protein-based (Ontak, Abraxane, Kadcyla, Pazenir, etc.) and metal-based (NanoTherm) ones, have progressed to clinical trials and were approved for clinical use [1,2,3]. However, the ability of our body to recognize and clear foreign material still remains a major issue for many nanotherapeutics showing promising results in preclinical studies. Biomimetic nanoparticles (BNPs) exhibit high biocompatibility and demonstrate the capacity to effectively traverse biological barriers [4,5]. Also, BNP design can be tailored using genetic and chemical methods, while new functionalization approaches can be endowed by fusion with organic and inorganic NPs [5].

Natural BNPs, e.g., exosomes or, more generally, extracellular vesicles (EVs), are produced by virtually all types of human cells [6,7]. EVs are secreted following intracellular generation via the inward budding of endosomal membranes in multivesicular bodies. They comprise a lipid bilayer enriched with various molecules (proteins, lipids, etc.) that define their tissue/cell-targeting properties and encapsulate a diverse array of biomolecules, including proteins, lipids and nucleic acids (typically, short RNAs), within their luminal compartment [8,9]. EVs represent a communication system enabling signal transduction in distant tissues and organs [10]. Discovered back in 1983 [11], they were proposed as a perspective delivery vehicle, first for miRNA and then for many other payloads (DNA, proteins, RNA, small molecules). Despite remarkable progress in the last decade, EVs continue to face challenges in their transition to clinical practice as delivery vehicles due to technological setbacks in their manufacturing and high heterogeneity that depend on multiple factors including cell source, cell type, and manufacturing conditions [12,13]. As an alternative, new types of BNPs were invented, such as exosome-mimetic nanovesicles (EMNVs) produced by extrusion of the whole cells through a series of filters with different diameters of pores, and nanoghosts fabricated from purified cell membranes [12,14,15]. Both types have clear advantages compared to EVs, such as a straightforward manufacturing process and markedly higher yields [16], while retaining the major properties of EVs, including biocompatibility, programmability, etc. [14,15,17]. While their loading with specific cargo can be challenging, one approach to achieve this is to fuse them with liposomes containing nucleic acids. Indeed, such approaches enable efficient packaging of RNA and DNA. Furthermore, hybrids of BNPs and liposomes demonstrate superior stability. When formulated with cationic lipids, these hybrids also exhibit enhanced loading efficiencies and transfection capabilities [18].

Internalization of most NPs, including BNPs, occurs through endocytosis followed by their entrapment and degradation in the endolysosomal compartment. However, there is evidence that biological NPs to some extent can enter human cells by direct membrane fusion with subsequent release of their constituents [19,20], therefore overcoming the endolysosomal compartment. The efficient escape of NPs from the endolysosomal compartment is the crucial step required for the release of cargo and the ultimate efficiency of the nanoformulation [21]. In general, the endolysosomal compartment consists of early endosomes maturing into late endosomes and, finally, fusion with lysosomes [22]. These organelles possess varying acidic pH levels (ranging from pH 6.5 to 4.5) necessary for their proper functioning and the activation of specific hydrolytic enzymes [22,23]. The degradation of payloads due to confinement within the endolysosomal compartment is well-documented [24]. Therefore, carrier structures are often designed to include cationic molecules, such as lipids and polymers enriched in amino groups, to disrupt the endosomal membrane and facilitate payload release.

In this study, for the first time, we investigated the behavior of all major types of biological BNPs (EVs, EMNVs, NGs) and hybrid BNPs in response to protonation. To accomplish this, we investigated BNP’s characteristics at different pH conditions. Our findings reveal the enhanced EMNV’s protonation compared to EVs and NGs. Furthermore, we demonstrate that incubating hybrid NPs in acidic pH for 15 min induces their rupture and results in the release of their constituents. Exposure to an acidic pH environment for a duration of 60 min led to disruption of EMNVs and hybrid NPs, with a more pronounced effect on the latter. In human cells, EMNVs mostly avoid the endolysosomal compartment, possibly to due previously observed fusion of biological NPs with the cell membrane. In contrast, hybrid NPs enter the endolysosomal compartment and escape it over time, consistent with its behavior in acidic pH. 

## 2. Materials and Methods

### 2.1. Cell Culture

HEK-293T and HepG2 cells were cultured in DMEM (PanEco, Moscow, Russia) supplemented with 10% fetal bovine serum (FBS) (Cytiva, Logan, UT, USA), 2 mM of L-glutamine, 100 U/mL of penicillin, and 100 µg/mL of streptomycin (Gibco, Thermo Fisher Scientific, Oxford, UK) at 37 °C and 5% CO_2_. 

### 2.2. Isolation of EVs

EVs were isolated from HEK-293T cell complete cell culture media as described previously [25]. Before isolation, HEK-293T cells were cultured in complete DMEM media supplemented with FBS depleted of animal EVs by ultrafiltration using Amicon Ultra-15 100 kDa filter devices (Merck Millipore, Darmstadt, Germany), as shown in [26]. In brief, conditioned media were centrifuged at 2000× *g* for 10 min and then at 10,000× *g* for 10 min to remove cell debris and large EVs followed by anion-exchange chromatography using MacroPrep DEAE Media anion-exchange resin (BioRad, Hercules, CA, USA). The column was washed with 100% buffer A (50 mM HEPES, 100 mM NaCl) followed by stepwise washing with 95% buffer A/5% buffer B (50 mM HEPES, 2 M NaCl)—5 column volumes (CV), 90% buffer A/10% buffer B—10 CV. The fraction containing EVs was eluted with 60% buffer A/40% buffer B and the column was then washed with 100% buffer B. The eluate was concentrated using Amicon Ultra-15 (100 kDa) filter devices followed by 3 washes with PBS and 1 more concentration step.

### 2.3. Preparation of EMNVs

HEK-293T cells were washed twice with PBS and detached using Versene solution (PanEco, Moscow, Russia). The cell suspension was serially extruded 7 times through 10 µm, 5 µm, and 1 µm polycarbonate membrane filters (Nuclepore, Whatman, Inc., Clifton, NJ, USA) using a mini-extruder (Avanti Polar Lipids, Birmingham, UK). The resulting solution was centrifuged for 10 min at 2000× *g*, then for 10 min at 10,000× *g* to discard cell debris; vesicles from the supernatant were isolated using Centrisart 1 (300 kDa MWCO) centrifugal concentrator (Sartorius, Goettingen, Germany) followed by 3 washes with PBS. mCherry protein was packaged into EMNVs and EMNVs-lipo using patented technology.

### 2.4. Preparation of NGs

HEK-293T cells were harvested, washed with PBS, re-suspended in cold TM buffer (0.01 M Tris, 0.001 M MgCl2, pH 7.4) and incubated for 20 min at 4 °C. Cell suspension was sonicated (5 s, 27% amplitude), and then mixed with 60% sucrose solution to a final concentration of 0.25 M of sucrose. The suspension was centrifuged at 6000× *g* for 15 min at 4 °C and the pellet was washed twice with 0.25 M sucrose in TM buffer (pH 7.4). The re-suspended pellet was sonicated (5 s, 27% amplitude) and centrifuged at 6000× *g* for 20 min at 4 °C. The pellet was washed twice with 0.25 M sucrose in TM buffer (pH = 8.6). The pellet was resuspended in PBS and consequently extruded through 1 µm (15 times), 400 nm (9 times) and 100 nm (15 times) polycarbonate membrane filters (GVS Filtration Inc., Bologna, Italy) using a mini-extruder (Avanti Polar Lipids, Birmingham, UK).

### 2.5. Preparation of Hybrid NPs

Liposomes were prepared by mixing 2.6 µg of RNA and 3.375 µL of Lipofectamine 3000 (Thermo Fisher Scientific, Oxford, UK) in 250 µL of Opti-mem media followed by 10 min incubation at room temperature. In total, 250 µL of EVs, NGs and EMNVs (1 × 10^11^ NPs/mL) was added to liposomes solution and gently mixed followed by incubation at 37 °C (20 min). Hybrids were extruded through 100 nm polycarbonate membrane filters (GVS Filtration Inc., Bologna, Italy) using a mini-extruder (Avanti Polar Lipids, Birmingham, UK). Unbound RNA was removed by ultrafiltration using Amicon Ultra-0.5 (100 kDa) filter devices (Merck Millipore, Burlington, MA, USA) followed by 3 washes with PBS.

### 2.6. Cryo-TEM

A lacey carbon-supported copper grid (200 mesh) was treated with air plasma to make it hydrophilic. Then, 3 µL of the sample was placed onto the treated grid. The excess of the sample was removed by blotting the grid for 1 s and then the grid immediately plunged into liquid ethane (automated plunging system, Vitrobot FEI, Hillsboro, OR, USA). Then, the grid with the sample was transferred into liquid nitrogen to the transmission electron microscope (TEM Tecnai G212 SPIRIT, FEI, Hillsboro, OR, USA).

### 2.7. Total Protein Measurements

Total protein content of BNPs was determined using the BCA Protein Assay Kit (FineTest, Wuhan, China) according to the manufacturer’s instructions. Absorbance was measured at 562 nm by Nano-500 (Allsheng, Hangzhou, China). Three biological replicates were measured for each analysis. 

### 2.8. Total Lipids Measurements

The total lipid content of BNPs was measured using sulfophosphovanilin assay as described previously [27]. Briefly, lipid standard solutions (2 μg/μL) were prepared from Dope (Sigma-Aldrich, St. Louis MO, USA) in chloroform and diluted from 100 to 0 μg per tube. In total, 25 μL of BNP isolate (150–300 μg/mL of protein) was added to the empty chloroform-pretreated Eppendorf tubes and 25 μL of PBS was also added to each lipid standard tube. Next, 125 μL of 96% sulfuric acid was added and incubated with open lids at 90 °C for 20 min. In total, 110 μL of samples and standards were pipetted into a 96-well plate. After cooling to room temperature, 55 μL of 0.2 mg/mL vanillin in 17% phosphoric acid was added to each well and shaken. Absorbance was measured at 540 nm by SuPerMax 3100 Multi-Mode Microplate Reader (Good Science, Tianjin, China). Three biological replicates were measured for each analysis. 

### 2.9. Western Blotting

EVs were lysed using 50 μL of RIPA buffer (50 mM Tris-HCl [pH 8.0], 150 mM NaCl, 0.1% Triton X-100, 0.5% sodium deoxycholate, 0.1% sodium dodecyl sulphate [SDS], 1 mM NaF) and incubated with agitation for 30 min at 4 °C followed by sonication for 30 s. Samples were loaded with 6× Laemmli buffer (5:1, 60 µg/well) onto 10% SDS-PAGE and then transferred on nitrocellulose membrane. The membrane was blocked with 5% milk in TBS-T (20 mM Tris, pH 7.5, 150 mM NaCl, 0.1% Tween 20) and stained with primary antibodies to exosome markers (EXOAB-KIT-1 for CD63, CD9, CD81 and Hsp70, SBI) 1:1000 or to β-actin (A1978, Sigma) 1:5000 in 5% milk in TBS-T overnight at 4 °C. Membranes were washed 3× with TBS-T and incubated for 1 h with goat anti-rabbit HRP-conjugated antibodies (Ab6721, Abcam, Cambridge, MA, USA) or with goat anti-mouse HRP-conjugated antibodies (Ab6787; Abcam) diluted 1:5000 in 5% milk in TBS-T. Membranes were washed 3 times with TBS-T and the chemiluminescent signal was developed with SuperSignal West Femto Maximum (Thermo Fisher Scientific, Oxford, UK) and detected with an X-ray film with 2 h exposure.

### 2.10. Nanoparticle Tracking Analysis (NTA)

NPs suspensions were analyzed using a Nanosight LM10 HS unit (NanoSight Ltd., Amesbury, UK) equipped with a 405 nm laser to determine the size and quantity of isolated particles after incubation in a buffer with pH 5.0–10.0. Videos of particle tracking were recorded at room temperature with passive temperature readout and the following camera setups optimized for EVs: camera shutter 1500, camera gain 500, lower threshold 195, and higher threshold 1885. The videos were processed with Nanoparticle Tracking Analysis analytical software version 2.3 (NanoSight Ltd., Amesbury, UK) with a detection threshold of 8 multi. At least 12 individual videos, each lasting 60 s, containing a minimum of 2000 tracks were recorded and processed [28]. Data from multiple videos were joined together to obtain a particle size histogram and the mean total concentration was corrected for dilution factor.

### 2.11. Dynamic Light Scattering (DLS)

A Malvern Zetasizer NanoZS instrument (Malvern, Worcestershire, UK) was used for the dynamic light scattering (DLS) analysis of all biological and hybrid biological nanoparticles. Each NP preparation was diluted 1/1000 in PBS with different pH (pH = 4.0–10.0) filtered through a 0.2 µM filter (Corning, New York, NY, USA) and analyzed at least 5 times; 1.5 mL of diluted preparations were loaded into polystyrene cuvettes (DTS0012; Malvern). Analyses were performed at 25 °C (100 measurements) using a 20 mW helium/neon laser (633 nm). Data were analyzed in Zetasizer software 8.01.4906 (Malvern Panalytical Ltd., Malvern, UK). Ζ-potentials were analyzed in U-type cuvettes (DTS1070; Malvern Panalytical Ltd., Malvern, UK) with gold electrodes at pH 7.4 and 4.0. Measurements of Ζ-potentials were performed at 25 °C at least 5 times. The background signal was measured in filtered PBS.

### 2.12. Native PAGE

In total, 20 µL of EMNVs and EMNVs-Lipo loaded with mCherry protein were mixed with 1.62 μL of 2 M NaOAc (pH 4.6) or with PBS (pH 7.4) and incubated for 15 min or 60 min at room temperature. Next, the samples were mixed with 4× sample loading buffer (125 mM Tris-HCl, pH 6.8, 20% glycerol (*v*/*v*), 0.004% bromophenol blue (*w*/*v*)). The proteins were separated at a 14% native PAGE and analyzed using the Bio-Rad ChemiDoc MP Imaging system with an Alexa546 filter. The intensity of the mCherry protein bands was determined using GelAnalyzer 19.1 software.

### 2.13. Endolysosomal Escape Analysis

HepG2 cells were treated with LysoTracker Green DND-26 dye at a concentration of 1 µM for 2 h. EMNVs or EMNVs-Lipo loaded with mCherry protein were added into the LysoTracker-containing cell culture medium for 2 h; then, cells were rinsed with phosphate-buffered saline (PBS) and supplemented with a new complete cell culture medium. The fluorescence intensity and spatial distribution of the LysoTracker dye were analyzed using an Image LCI Image ExFluorer Microscope equipped with EGFP and Texas Red filters and a module of artificial intelligence. Co-localization analysis was performed by quantifying the number of overlapping green and red pixels within the cells. At least 5000 cells were taken into analysis. 

### 2.14. Colocalization Analysis

Colocalization analysis was performed in Image ExFluorer Software (01 Image ExFluorer, Seoul, Republic of South Korea). Pearson and Mander colocalization coefficients were calculated in ImageJ (Version 1.54i) using the JACoP plugin.

### 2.15. Statistical Analysis

Values were expressed as the mean ± standard deviation (SD) in GraphPad Prism 8.0 software. Data were compared using parametric analysis of variance (ANOVA) or paired *t*-test and calculated *p*-values to determine statistically significant differences in means. 

## 3. Results

### 3.1. Isolation, Production and Characterization of EVs, NGs and EMNVs

First, we produced three types of BNPs (EVs, NGs and EMNVs) and performed characterization of their physical, chemical and biological properties. Cryo-TEM confirmed the generation of spherical NPs that appeared different in electron density and had spherical shape (Figure 1A–C). NTA analysis revealed that EMNVs and NGs had a modal size of ~60–100 nm, while EVs were more heterogeneous (Figure 1D–F). The observed differences in the size of EVs compared to EMNVs and NGs were expected considering the natural origin of EVs compared to the synthetic origin of EMNVs and NGs generated through multiple and consistent extrusion steps. EVs were additionally characterized by Western blotting as required by MISEV [29], demonstrating the expression of common exosome markers (CD9, CD63, CD81, Hsp70) and low levels of β-actin (Appendix A). 

### 3.2. Fabrication of Hybrid NPs

Next, we fabricated hybrid NPs by fusing liposomes with EVs, EMNVs and NGs. Hybrid NPs are perspective delivery vehicles that combine the advantages of biological NPs (high biocompatibility, ability to cross biological barriers, programming, etc.) with liposomal formulations enabling loading of high amounts of therapeutics RNAs (e.g., siRNA, shRNA, etc.) [18] and increasing penetration of NPs into hard-to-transfect cells in vitro and in vivo [30]. Ample studies demonstrated that fusing EVs with liposomes and cationic lipids increased the efficiency of cargo delivery [30,31]. 

As fusing liposomes with biological NPs increases their overall size over the optimal range of ~100 nm, required for efficient in vivo delivery [32], we generated hybrid NPs using a two-step protocol including (1) fusion with liposomes and (2) extrusion of obtained hybrid BNPs through 100 nm pore membranes. As a result, produced hybrid BNPs were very uniform in shape (Figure 2A–C) and had highly homogeneous size distribution with a modal size of ~100–120 for all three types of BNPs, including originally heterogeneous EVs (Figure 2D). Expectedly, fusion of BNPs with liposomes resulted in increased lipid-to-protein content of hybrid NPs (Figure 2G–I).

### 3.3. Effects of pH on Size and ζ-Potential of Biological and Hybrid NPs

After cell internalization, BNPs are sequestered in the endosomal acidic compartment. This is one of the major biological barriers, and their stability in these conditions as well as their ability to escape the endolysosomal compartment define their overall efficiency as delivery vehicles [33]. The incorporation of cationic lipids into the composition of hybrid NPs increases endosomal escape upon endo-lysosomal maturation and acidification via a proton sponge effect [34]. The proton sponge effect is a hypothetical mechanism whereby weakly basic NPs absorb free protons, resulting in the influx of protons and counter ions, changing the pH and causing the rupture of endosomes and the release of cargo [35].

To mimic endolysosomal entrapment of BNPs and hybrid BNPs in vitro and investigate the effect of protonation on their properties (size, size distribution and ζ-potential), we incubated them in acidic pH (4.0) and physiologic condition with pH (7.4). Expectedly, acidic pH increased the surface charge of all the analyzed particles, reaching significant values for EVs (*p* < 0.01) and EMNVs (*p* < 0.001) and hybrid BNPs (*p* < 0.0001) (Figure 3). 

On the other hand, DLS revealed that in acidic conditions the size of EVs and NGs was unaltered, whereas EMNVs significantly increased their size from ~130 nm to ~230 nm (*p* < 0.0001) (Figure 4A–F), indicating substantial differences in the surface characteristics of EMNVs and EVs/NGs. 

The size increase of hybrid NPs to acidic pH was even more pronounced with all studied BNPs (EVs-Lipo: ~100 nm to ~130 nm; NGs-Lipo: ~140 nm to ~250 nm; EMNVs-Lipo: ~90 nm to ~280 nm) (Figure 4A–F). Size distribution analysis further indicated that the observed increase in the mean size of EMNVs and hybrid NPs was associated with the overall enlargement of NPs. The observed increase in the size of NPs can be explained by the influx of ions and fluid into the lumen of NPs resulting in the “swelling” of NPs, an important indication of the sponge effect mechanism. 

Collectively, these data demonstrate a more pronounced response of EMNVs to acidic pH compared to EVs and NGs, and a markedly higher response of hybrid NPs. Increased protonation and swelling of EMNVs and, especially, hybrid NPs, point to a previously unknown mechanism promoting endolysosomal escape and better cargo delivery of EMNVs and hybrid NPs observed in published studies. 

### 3.4. Mimicking Entrapment of NPs in Lysosomes Reveals Rupture of Hybrid and, to a Lesser Extent, Biological NPs

Next, chose EMNVs and EMNVs-Lipo to study the values of pH when changes in size and charge of NPs occur. EMNVs were chosen for these experiments due to their more pronounced response to acidic pH. While cytoplasmic pH is neutral (pH = 7.4), the pH of the endosomal and lysosomal lumen varies from 4.5 to 6.5 from more neutral in early endosomes (pH = 6.5) to acidic in late endosomes (pH = 5.5) and highly acidic in lysosomes (pH = 4.5–5.0) [36]. Alkalic solutions (pH = 8.0 and 10.0) were used as controls. 

For both EMNVs and EMNVs-Lipo, the mean size of NPs increased only at acidic pH values corresponding to late endosomes and lysosomes (pH = 5.0), suggesting that swelling of NPs may occur progressively in late endosomes (pH = 5.0) and lysosomes (pH = 4.0–4.5), but not in early endosomes (pH = 6.5) (Figure 5A,B). Protonation of either EMNVs or EMNVs-Lipo did occur in acidic pH = 5.0 and pH = 6.5. Alkaline conditions did not alter the size of NPs and gave EMNVs, but not EMNVs-Lipo, a more negative charge (Figure 5). 

Next, we compared the size distribution and stability of EMNVs and EMNVs-Lipo at pH = 4.0 and pH = 7.4 using NTA, showing that at acidic pH, the number of EMNVs does not change, whereas in the case of EMNVs-Lipo, it dramatically declines (Figure 6A). CryoTEM analysis revealed swelling and rupture of EMNVs-Lipo in acidic pH (Figure 6B). To further test the rupture of EMNVs-Lipo, we loaded EMNVs and EMNVs-Lipo with a fluorescent protein mCherry and measured values of mCherry fluorescence using a native PAGE (Figure 6C,D). Using this test, if NPs are not ruptured, the mCherry signal remains in the wells of the gel as intact NPs do not run in the gel due to very high molecular weight. In contrast, ruptured NPs would have released the luminal mCherry protein that can run on the gel at a molecular weight of 35–55 kDa. When such NPs were run on PAGE, mCherry was still detected at both pH = 7.4 and pH = 4.0, indicating that NPs per se release mCherry in these conditions. However, at pH = 7.4, substantial amounts of mCherry remain within NPs and stay at a weight of >250 kDa. Exposure of either type of NPs to acidic pH releases virtually all mCherry protein trapped in NPs. 

Incubating EMNVs for 15 min at acidic pH does not change mCherry release, but longer incubation for 60 min significantly increases mCherry release (Figure 6C,D). At the same time, EMNVs-Lipo are ruptured already during 15 min long incubation at pH = 4.0 with a dramatic increase in mCherry fluorescence. Notably, it appears that EMNVs-Lipo are ruptured at a very short period of time, as longer incubation in acidic pH does not further increase mCherry release (Figure 6C,D).

These data indicate that EMNVs are ruptured in acidic pH during incubation at pH = 4.0 for 60 min, but less pronouncedly during incubation for 15 min. At the same time, EMNVs-Lipo are all ruptured during 15 min incubation. Overall, it points to the higher ability of hybrid NPs to execute the proton sponge mechanism of endolysosomal escape. By increasing the binding of protons and enhancing water ingress, the size of NPs is augmented, leading to NP rupture and subsequent release of proteins. 

### 3.5. EMNVs Escape Endolysosomal Compartment More Efficiently Than Hybrid EMNVs-Lipo

The above experiments demonstrated how BNPs and hybrid NPs behave in an acidic pH environment corresponding to different endolysosomal compartments, revealing that EMNVs and hybrid NPs could be disrupted by releasing their cargo in highly acidic conditions. Next, it was important to understand whether these properties would provide better endolysosomal escape of NPs in human cells. To test this, we stained the endolysosomal compartment in human HepG2 cells with Lysotracker dye and loaded EMNVs and hybrid EMNVs-Lipo with mCherry protein. At 3 h after adding mCherry-loaded NPs to HepG2 cells, NPs were washed out, and HepG2 in FluoroBright complete medium was visualized for the next 36 h. Fluorescent time-lapse microscopy demonstrated that EMNVs are markedly more efficient in avoiding the endolysosomal compartment compared to hybrid NPs (Figure 7A–C). Notably, from the first timepoint, EMNVs were mostly not colocalized with the endolysosomal compartment, and this colocalization efficiency had not changed over time (Figure 7D–G), pointing at preferential avoidance of endolysosomal organelles. At the same time, analysis of hybrid NPs showed endosomal escape during analysis, as colocalization of hybrid NPs decreased by the end timepoint (Figure 7D–G), indicating efficient escape from the endolysosomal compartment which is consistent with their behavior in acidic pH. Similarly, hybrid NPs markedly increased their escape when double the amount of NPs was added to HepG2 cells (Figure 7D–G), which supposedly corresponds to better escape by the proton sponge mechanism. 

## 4. Discussion

BNPs are perspective delivery vehicles with unique characteristics of biocompatibility, low to no immune response and reduced clearance by macrophages, safety, functionalization properties and ability to cross biological barriers [37]. EVs have already been used to deliver therapeutic RNA, gene editing CRISPR/Cas complexes, small molecules, viruses, plasmid DNA, etc. [38,39,40,41]. EVs are under clinical investigation for therapy of autoimmune diseases, in regenerative medicine, etc. [3,42,43]. However, EVs are hard to manufacture, and their production is fraught with low yields of NPs, high heterogeneity, complex isolation, purification and characterization, making the medical-grade EV production process hard to meet the GMP’s standards while skyrocketing the costs [12]. The field of EMNVs and NGs is relatively new but is rapidly gaining ground in novel drug nanoformulations. EMNVs/NGs retain the advantages of EVs but have a simple and straightforward manufacturing pipeline ensuring yields of NPs exceeding hundreds to thousands of times that of EVs [12,44]. 

The internalization routes and efficiency of NPs are influenced by various factors such as size, shape, charge, stiffness, hydrodynamic volume, and materials of the NPs, along with additional constituents that promote interactions with cell membranes [45]. The nature and characteristics of acceptor cells also strongly affect the internalization routes and fate of internalized NPs, as different cell types and cancers exhibit variations in the structure and composition of their cell membranes [46]. The pH in the cytoplasm of cancer cell types deviates from physiological values, impacting the entry and cargo release of NPs [47]. Interaction of NPs with serum or other biological fluids can lead to protein corona formation, altering the size, charge, and surface properties of the NPs [48]. Size and surface charge play important roles in determining the cellular internalization route. Previous studies suggest that NPs within a size range of 50–300 nm are mainly internalized through clathrin-mediated and caveolin-dependent endocytosis. NPs smaller than 50 nm are internalized via caveolin-mediated endocytosis, with micropinocytosis serving as an auxiliary pathway under specific conditions [49]. Macropinocytosis is observed for NPs larger than 250 nm. Surface charge influences the internalization of NPs through different endocytosis pathways. Cationic NPs (>+10 mV) with sizes between 50 nm and 200 nm exhibit strong electrostatic interactions with cells, leading to rapid internalization primarily through clathrin-mediated endocytosis, as well as caveolae, micropinocytosis, and clathrin/caveolae-independent routes. Conversely, small NPs (<50 nm) with surface charges ranging from +15 to −15 mV utilize the caveolin-mediated route. Anionic NPs can be endocytosed through caveolae-mediated pathways in the presence of positively charged components in their membranes. Neutral NPs are internalized based on hydrophobic and hydrogen bonding interactions [45]. 

The current state-of-the-art suggests the existence of several mechanisms for overcoming the endolysosomal barrier, such as the proton sponge effect, osmotic lysis effect, swelling effect, pore formation, membrane disruption, membrane fusion, and photochemical internalization. Alternatively, certain types of NPs can avoid the endolysosomal compartment by directly fusing with the plasma membrane, utilizing caveolae-mediated transport, or undergoing direct transport to the Golgi and endoplasmic reticulum [50]. BNPs reportedly can fuse with plasmalemma and endosomal membranes releasing cargo directly into the cytoplasm, but the major route of internalization occurs via multiple endocytic pathways, mostly endocytosis [21]. EVs can be internalized through endocytosis, ligand–receptor interactions, or direct fusion with the cell membrane. EMNVs undergo internalization via similar routes but demonstrate consistently higher efficiency [40]. A head-to-head comparison of EVs and EMNVs also reveals differences in homologous (delivery of NPs to cells of the same origin) and heterologous (to other cell types) cellular uptake. For homologous cells, EV uptake occurs via caveolin-independent endocytosis, while EMNVs are internalized through micropinocytosis. In the case of heterologous cells, internalization is clathrin- and caveolin-dependent for both EVs and EMNVs [51]. Fusion of EVs with liposomes results in the formation of EV–liposome hybrids, which exhibit higher internalization rates compared to EVs and demonstrate lower toxicity than liposomes [18]. The fate of endocytosed entities depends entirely on whether they escape degradation in the endolysosomal pathway or undergo decay by lysosomal enzymes in an acidic environment [52]. To date, natural mechanisms of endolysosomal escape of biological NPs are poorly investigated. There are several putative mechanisms, including (1) fusion of biological NPs with endo/lysosomal membranes, (2) proton sponge mechanism, (3) swelling of pH-sensitive NPs and (4) destabilization of endo/lysosomal membranes [53]. Incorporation of cationic lipids or polymers enriched in amine groups induces protonation of NPs and endo/lysosomal escape via the proton sponge effect, associated with counterions of Cl^−^ and influx of water. Increasing salt concentration creates osmotic pressure that swells and ruptures endosomes with a subsequent release of their contents [50,54].

Interestingly, previous studies did not attempt to analyze the behavior of biological and hybrid NPs in acidic conditions upon interaction with an acidic environment or during short-term incubation. Instead, several studies focused on the effects of different pH on the storage of EVs [55]. In our study, for the first time, we directly compared the interaction of all major types of biological and hybrid NPs with acidic solutions, demonstrating their protonation and swelling evidenced by a moderate increase in the mean size of NPs (Figure 3, Figure 4 and Figure 5). EMNVs were more prone to protonation and swelling compared to EVs and NGs. This observation suggests another mechanism of increased cargo delivery by EMNVs shown in previous studies [30]. Incubation of NPs in acidic pH conditions corresponding to early and late endosomes and lysosomes demonstrated that quantities of hybrid NPs are markedly reduced, suggesting that hybrid NPs are effectively ruptured in conditions of late endosomes and lysosomes (Figure 6). This was further confirmed by native PAGE with mCherry-loaded EMNVs and EMNVs-Lipo which demonstrated that EMNVs-Lipo are rapidly ruptured at pH = 4.0 releasing mCherry protein, whereas EMNVs per se are effectively ruptured only after incubation at pH = 4.0 for 1 h. These demonstrate that fusion of biological NPs with liposomes markedly increases their propensity for executing the proton sponge effect and releasing vesicular cargo into the cytoplasm. At the same time, generating hybrid NPs from EMNVs markedly changes their internalization route, as EMNVs per se mostly avoid the endolysosomal compartment as evidenced by timelapse fluorescent analysis (Figure 7), while hybrid EMNVs fused with liposomes enter the endolysosomal compartment and escape it over the observation period from 6 to 36 h. This suggests that (1) the proton sponge effect plays a minor role in the endolysosomal escape of EMNVs, but not hybrid EMNVs, and (2) direct fusion may represent the predominant mechanism by which biological EMNVs avoid the endolysosomal compartment. 

## 5. Conclusions

EMNVs are more susceptible to protonation and swelling as compared to EVs and NGs. Fabrication of hybrid NPs composed of biological NPs and liposomes enhances their abilities to execute the proton sponge mechanism of endolysosomal escape. EMNVs predominantly avoid the endolysosomal compartment, whereas hybrid EMNVs-Lipo escape it, likely due to the proton sponge

## Figures and Tables

**Figure 1 pharmaceutics-16-00667-f001:**
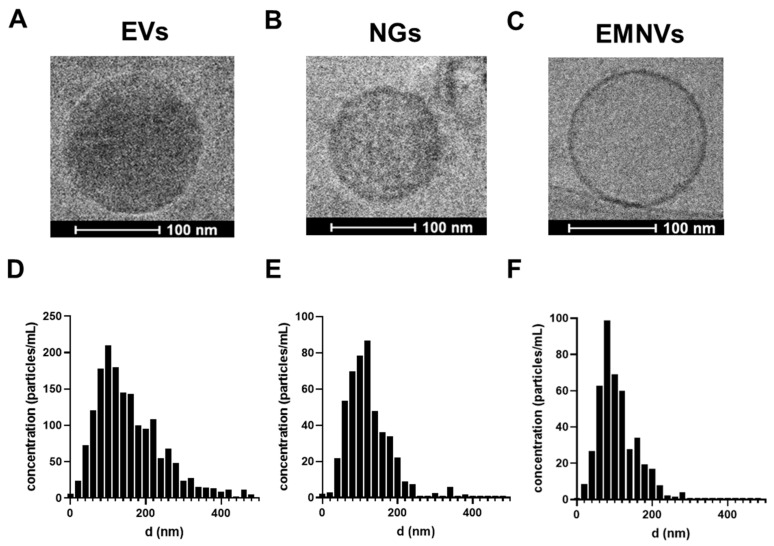
Characterization of NPs. Cryo-TEM images of (**A**) EVs, (**B**) NGs and (**C**) EMNVs. Nanoparticle tracking analysis of NPs: (**D**) EVs, (**E**) NGs and (**F**) EMNVs.

**Figure 2 pharmaceutics-16-00667-f002:**
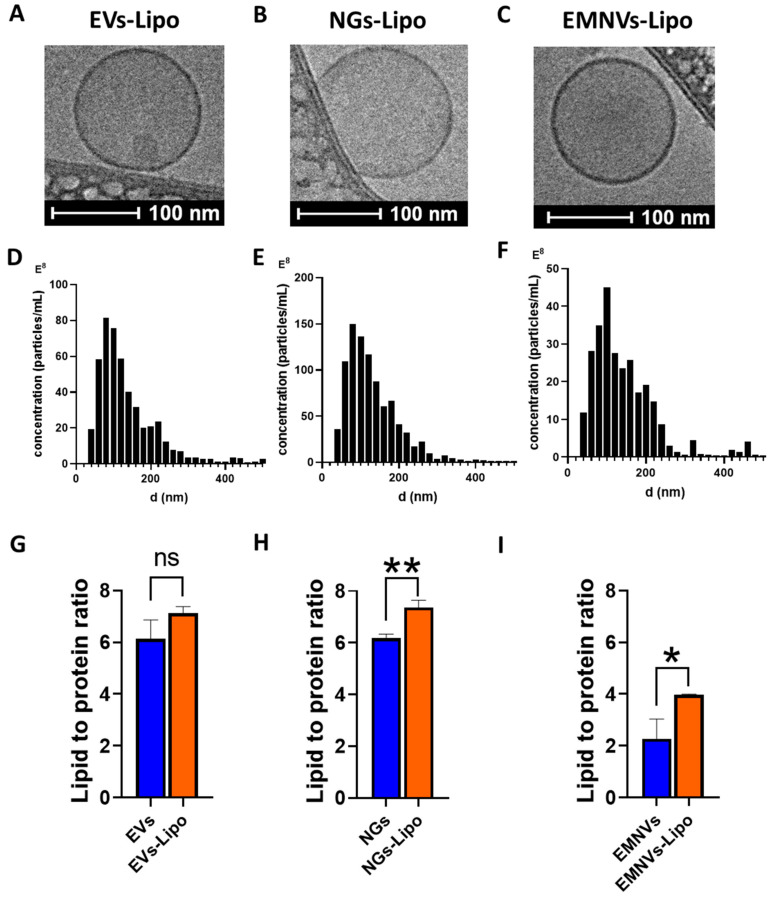
Characterization of hybrid NPs. Cryo-TEM images of (**A**) EVs-Lipo, (**B**) NGs-Lipo and (**C**) EMNVs-Lipo. Nanoparticle tracking analysis of NPs: (**D**) EVs-Lipo, (**E**) NGs-Lipo and (**F**) EMNVs-Lipo. Lipid-to-protein ratio measured for (**G**) EVs and EVs-Lipo, (**H**) NGs and NGs-Lipo and (**I**) EMNVs and EMNVs-Lipo. * *p* < 0.05, ** *p* < 0.01. ns—non significant. Three replicates were collected for each sample to quantify protein and lipid concentrations.

**Figure 3 pharmaceutics-16-00667-f003:**
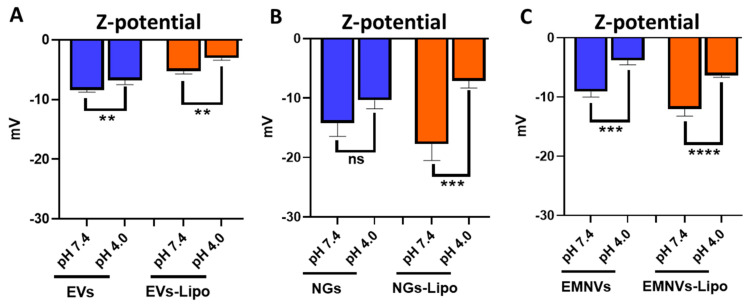
Dynamic light scattering analysis of NPs/NPs-Lipo at pH 7.4 and pH 4.0. Ζ-potential of (**A**) EVs/EVs-Lipo, (**B**) NGs/NGs-Lipo and (**C**) EMNVs/EMNVs-Lipo. Each type of NPs was analyzed at least 5 times. ** *p* < 0.01, *** *p* < 0.001, **** *p* < 0.0001, ns—not significant.

**Figure 4 pharmaceutics-16-00667-f004:**
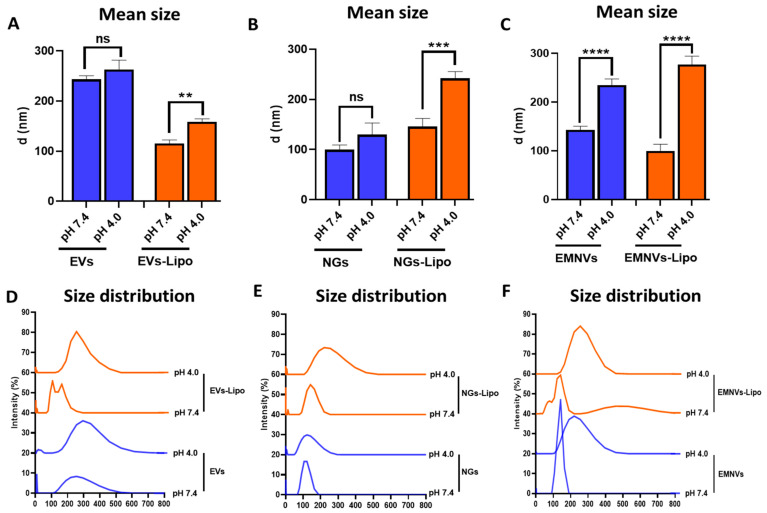
Dynamic light scattering analysis at pH 7.4 and pH 4.0. (**A**–**C**) Mean diameter and (**D**–**F**) size distribution of biological and hybrid NPs. Each type of NPs was analyzed at least 5 times. ** *p* < 0.01, *** *p* < 0.001, **** *p* < 0.0001, ns—not significant.

**Figure 5 pharmaceutics-16-00667-f005:**
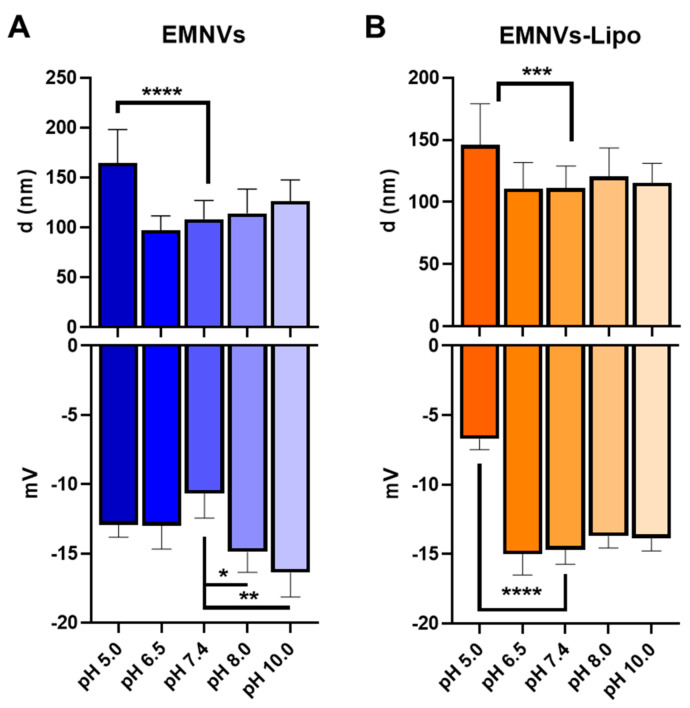
Average size and Z-potential of EMNVs/EMNVs-Lipo at different pH. (**A**) Mean size and ζ-potential of EMNVs, (**B**) Mean size and ζ-potential of EMNVs-Lipo. Each type of NPs was analyzed by NTA (for mean size measurements) and at least 5 times by DLS (for ζ-potential measurements). * *p* < 0.05, ** *p* < 0.01, *** *p* < 0.001, **** *p* < 0.0001.

**Figure 6 pharmaceutics-16-00667-f006:**
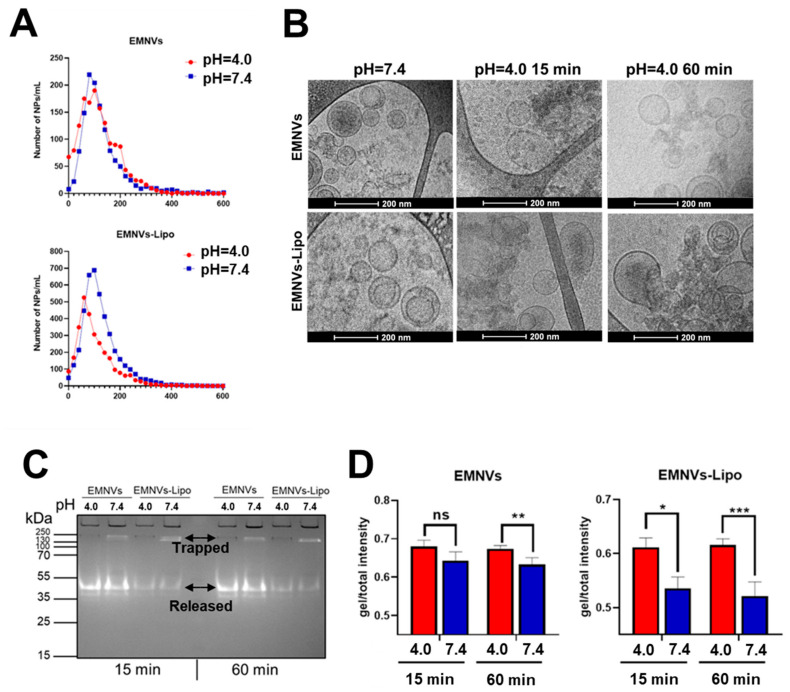
Rupture and release of vesicular cargo from EMNVs and EMNVs-Lipo. (**A**) Nanoparticle tracking analysis of EMNVs and EMNVs-Lipo incubated at pH 4.0 (red lines) and pH 7.4 (dark blue lines) for 15 min. (**B**) CryoTEM images of EMNVs and EMNVs-Lipo after incubation at different pH for 15 min and 60 min. (**C**) Native PAGE of EMNVs and EMNVs-Lipo containing mCherry protein at pH 4.6 and pH 7.4 after incubation for 15 and 60 min and (**D**) respective values. * *p* < 0.05, ** *p* < 0.01, *** *p* < 0.001, ns—not significant.

**Figure 7 pharmaceutics-16-00667-f007:**
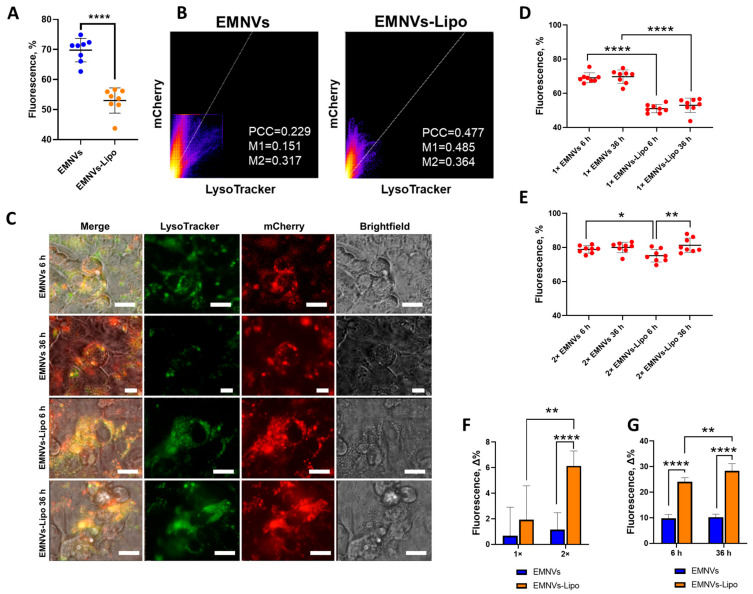
Endolysosomal escape of EMNVs and hybrid EMNVs-Lipo. HepG2 cells were stained with LysoTracker Green DND26 (green) and treated with mCherry-loaded EMNVs or EMNVs-Lipo (red). Efficiency of endolysosomal escape was measured as the percentage of the mCherry signal that does not colocalize with the green signal. (**A**) Percentage of mCherry-loaded NPs escaping the endolysosomal compartment presented as mean values for 8 images of at least 600 cells. (**B**) Colocalization analysis of mCherry/LysoTracker with EMNVs and EMNVs-Lipo. (**C**) Representative fluorescence images of HepG2 cells. Scalebar: 20 μm. Percentage of mCherry-loaded EMNVs or EMNVs-Lipo escaping the endolysosomal compartment at 6 h and 36 h after NPs treatment adding (**D**) 1× or (**E**) 2× concentration of NPs. (**F**) Change in endosomal escape of 1× or 2× concentration of NPs presented as the difference in cytoplasmic mCherry at 36 vs. 6 h. (**G**) Change in endosomal escape at 6 h and 36 h after adding NPs presented as a difference in cytoplasmic mCherry at 1× or 2× concentrations. * *p* < 0.05, ** *p* < 0.01, **** *p* < 0.0001.

## Data Availability

All data is contained within the article and Appendix A.

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
