# Peer review of "Swelling, Rupture and Endosomal Escape of Biological Nanoparticles Per Se and Those Fused with Liposomes in Acidic Environment"

_pharmaceutics, 2024, doi:10.3390/pharmaceutics16050667_

Round 1
Reviewer 1 Report
Comments and Suggestions for Authors
This is an interesting work presenting insights on the characteristics of types of biological biomimetic nanoparticles, their integration with liposome and following responses to acidic environments. This could assist in understanding the nanoparticles’ behaviour through endo/lysosome escape process, and help develop more effective nano-therapeutics carriers. The study is well designed with careful investigations. The result section of this work presented a clear logical flow from fabricating the nanoparticle, characterizing the nanoparticles when exposed to a low pH environment and evaluating the endosomal escape efficiency. I would recommend acceptance of this manuscript with minor revisions considering comments below.
- In the result section “3.2. Fabrication of hybrid NPs”, lines 285 to 286, “Expectedly, fusion of BNPs with liposomes resulted in increased lipid-to-protein content of hybrid NPs” but Figure 2G, shows no significant difference in lipid to protein ratio between EV to EV-liposomes. There seems to be trend comparing the three groups of BNPs, please provide possible analysis and explanations.
- It is interesting to see a systematic analysis of these BNPs, protein/lipid ratio, size, zeta potential, etc., and more importantly the comparisons between groups and stimuli (acid). However, what is the sample size for these statistical analysis? Please provide details on this information in the Experimental Section and each figure cation.
- The scale bar in Figure 7C in the result section 3.5 is way too small, it will be better to enlarge it.
- For Figure 7c, the confocal images are suggested to be taken under a higher magnification to show the endosome escape process clearly.
Author Response
Please, find the responses to the Reviewer's comments in the attached file.
We highly appreciate your work and valuable comments.

Reviewer 2 Report
Comments and Suggestions for Authors
This article focuses on comparing the mechanisms of phagocytosis and escape at the cellular level of 4 kinds of biological NPs, which are of high value for enhancing the understanding of the escape of biomimetic drugs from endolysosomal compartments, and therefore for the design of the biomimic drug carriers. The study is interesting and well designed. howevery,I have some comments that may improve the quality of the manuscripts.
1. the size distribution of 3 naive biological NPs their cresponding lipd hybird nanogost are different, would their size distribution affect their escpe of the endolysosomal compartments? please give some discussion
2. in figure 6B, only CryoTEM images of NPs incubated at various PH for 15 min was provided, it would be more convincing if the author could provid their CryoTEM micrographs at 60 minutes of incubation at various PH.
3. and another question in figure 6B, the representative CryoTEM images of EMNVs at pH= 4.0 seems ruptured even more than the EMNVs-lipo, which did not match the explanation in the text.
4. in figure 6c, the time point of th representative fluorescence images of HepG2 cells should be provided, and also should better give two fluorescence images at 6h and 36 h.
Author Response

(The authors gave the same response as above.)

Round 2
Reviewer 2 Report
Comments and Suggestions for Authors
The author has improved the quality of the manuscript,and the questions have been handled properly. however, i consider that the NTA description in figure 1, 2 and 5, "All samples were recorded and processed at least 12 times." should not be shown, 12 times seem a little strange since they are not traditional repeat with different sample. I think the information shown in the method section 2.10 is clear, suggesting that the 12 times are just analytical methods with one experiment, not 12 independent repetitions of the experiment.
Author Response
The sentence was deleted from Figures 1, 2 and 5.
We again thank the Reviewer for the time and effort taken for reviewing the manuscript.
